# Electron correlation driven non-adiabatic relaxation in molecules excited by an ultrashort extreme ultraviolet pulse

A. Marciniak[1], V. Despré[2], V. Loriot[1], G. Karras[1], M. Hervé[1], L. Quintard[3], F. Catoire[3], C. Joblin[4], E. Constant [1,3], A.I. Kuleff[2] & F. Lépine[1]

The many-body quantum nature of molecules determines their static and dynamic properties, but remains the main obstacle in their accurate description. Ultrashort extreme ultraviolet pulses offer a means to reveal molecular dynamics at ultrashort timescales. Here, we report the use of time-resolved electron-momentum imaging combined with extreme ultraviolet attosecond pulses to study highly excited organic molecules. We measure relaxation time-scales that increase with the state energy. High-level quantum calculations show these dynamics are intrinsic to the time-dependent many-body molecular wavefunction, in which multi-electronic and non-Born−Oppenheimer effects are fully entangled. Hints of coherent vibronic dynamics, which persist despite the molecular complexity and high-energy excitation, are also observed. These results offer opportunities to understand the molecular dynamics of highly excited species involved in radiation damage and astrochemistry, and the role of quantum mechanical effects in these contexts.

[1] Institut Lumière Matière, Université Lyon 1, CNRS, UMR 5306, 10 rue Ada Byron, 69622 Villeurbanne Cedex, France. [2] Theoretische Chemie, PCI, Universität Heidelberg, Im Neuenheimer Feld 229, 69120 Heidelberg, Germany. [3] Université Bordeaux, CEA, CNRS, CELIA, UMR 5107, 33400 Talence, France. [4] Institut de Recherche en Astrophysique et Planétologie IRAP, Université de Toulouse (UPS), CNRS, CNES, 9 Av. du Colonel Roche, 31028 Toulouse Cedex 4, France. Correspondence and requests for materials should be addressed to A.I.K. (email: alexander.kuleff@pci.uni-heidelberg.de) or to F.L. (email: franck.lepine@univ-lyon1.fr)

Thanks to the development of ultrafast spectroscopic techniques, the investigation of the dynamics of few-body atomic or molecular systems has led to the observation of mechanisms down to the attosecond timescale[1,2]. Such experiments have provided information on how nature functions at the quantum level[3,4] through the study of resonances[5–7], quantum coherences[8–10], and energy flow[11–13]. However, many-body quantum systems remain a major challenge[14] both for experiment and theory. The main difficulty in describing many-body systems is that the two conceptual pictures for structure and dynamics often break down. Indeed, the orbital picture describing the electronic molecular states as single electron configurations distributed in molecular orbitals becomes less adequate for highly excited states[15,16]. It can also be the driving force for nontrivial dynamics, such as ultrafast correlation-driven hole migration[17,18]. At the same time, with an increase of the density of electronic states, the Born−Oppenheimer approximation, which decouples the electronic and nuclear wavefunctions, may fail leading to the appearance of nonadiabatic relaxation through conical intersections (CIs)[19]. In this context, ultrashort extreme ultraviolet (XUV) pulses are a useful tool to reveal the dynamics of many-body systems, because they combine high-energy photons and short duration times. Indeed, XUV excitation may populate a multitude of non-Born−Oppenheimer states with strong multielectronic character, leading to a truly many-body molecular wavefunction whose time evolution is driven by the complex entanglement between the large number of electronic and nuclear degrees of freedom.

Here, we investigate naphthalene (Naph, $C_{10}H_8$) and adamantane (Ada, $C_{10}H_{16}$) using an XUV-pump−IR-probe scheme (Fig. 1a). The XUV pulse ionizes the neutral molecules and populates excited cationic states, with energies near the double-ionization level (IP2 = 21.5 eV) (Fig. 1b). The dynamics of these excited cations are probed with a second ionization, performed using a delayed IR pulse (800 nm, 1.55 eV). Hence, the time-resolved photoelectron spectrum (TR-PES) probes the dynamics of the many-body quantum cationic states (Fig. 1c). These dynamics can only be explained when both strong multielectronic[20] and non-Born−Oppenheimer[21] effects are taken into account.

## Results

**Angle resolved photoelectron XUV−IR pump−probe spectroscopy.** The experiment was performed on an XUV beamline consisting of a compact XUV-IR interferometer based on an XUV attosecond pulse train with an envelope of sub-30 fs and fs IR beams, coupled to a velocity map imaging (VMI) spectrometer (see Methods). The XUV pulse was synthesized via high harmonic generation (HHG). The experimental results were obtained with an HH spectrum containing harmonics between 17 and 35 eV, centered around 26 eV.

We recorded a statistical set of two-color photoelectron spectra as a function of the XUV-pump–IR-probe delay. The photoelectron spectrum was obtained after angular integration of the VMI image and the two-color signal defined as the difference between the signal measured when the two pulses are present and the signal measured with XUV only. By doing so, we subtract the contribution of electrons originating from the single and double ionization of the molecule by one XUV photon. Above an electron kinetic energy ($E_{kin}$) of 1 eV, the signal is symmetric in time and stems from the XUV ionization of the neutral Naph assisted by the IR near $t = 0$ (not shown). Below 1 eV, the signal is asymmetric in time and corresponds to an XUV-pump–IR-probe scheme. We thus focus our analysis on those photoelectrons below 1 eV that are produced upon IR ionization of the cationic Naph.

The time-dependent two-color TR-PES is plotted in Fig. 2a. In this map we distinguish three features, corresponding to three maxima in the spectrum: around 0.4, 0.64 and 0.88 eV, respectively. By fitting the signal at these energies integrated between ±0.075 eV (Methods) the extracted time-constants $\tau_{decay}$ are 24 ± 5, 33 ± 6, and 46 ± 7 fs, respectively (Fig. 2b). This leads to the main observation of this work, which is that the time-constant increases with the photoelectron energy indicating that the closer the cationic states are to the double-ionization threshold, the longer the dynamics are. We performed the same experiment on Ada that is a similar size carbon-based molecule but with a nonaromatic, 3D structure. Here again we observed that the decay timescale increases with the photoelectron energy (Supplementary Fig. 6) which demonstrates that this effect is not specific to Naph. We have also changed the XUV spectrum and IR intensity and no significant variation of the timescale was observed (Fig. 2d, f).

What kind of dynamics does this reflect? As shown in Fig. 3a, b, within the monoelectronic and Born−Oppenheimer approximation, one would expect three completely decoupled cationic states (corresponding to the ionization out of three inner-valence

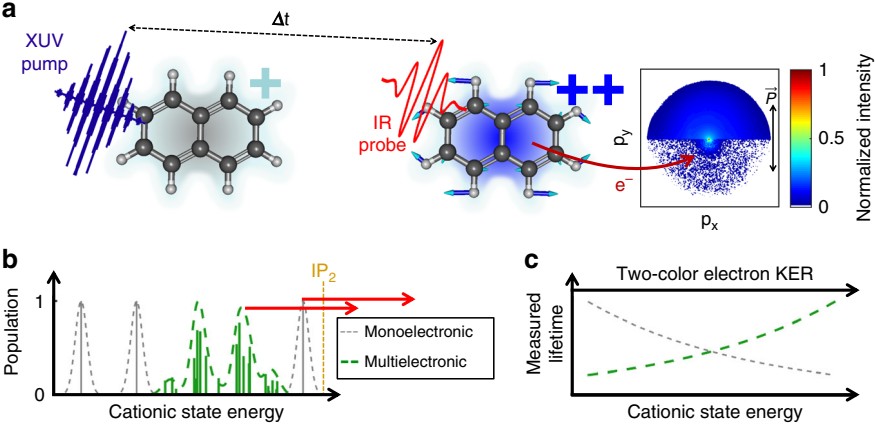

**Fig. 1** Ultrafast correlation-driven relaxation dynamics of naphthalene cation. **a** A short XUV pump pulse is followed by a time-delayed short IR probe pulse which interacts with the Naph (or Ada) molecules. The light polarization is set parallel to the detector. The emitted photoelectrons are collected with a velocity map imaging spectrometer, allowing measurement of the electron-momentum distribution as a function of the delay between pulses. **b** The cationic states produced by the XUV ionization have strong multielectronic character which leads to a complex structure in which multielectronic (in green) and vibrational degrees of freedom are all strongly coupled. **c** The IR pulse produces electrons that probe the dynamics of these states revealing an energy dependence of the lifetime intrinsic to the many-body character of the molecular states (green line)

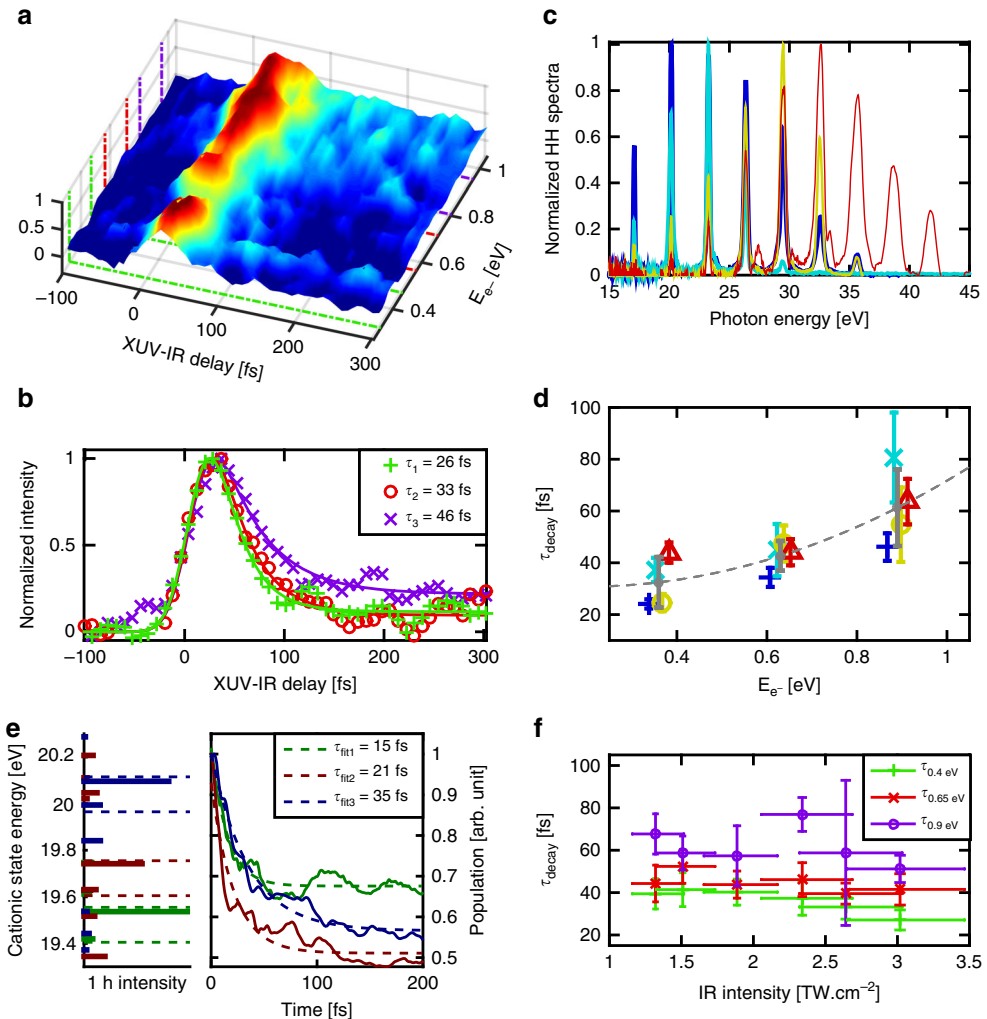

**Fig. 2** Ultrafast relaxation dynamics of naphthalene cation. **a** Time-dependent two-color electron kinetic energy spectrum for Naph. Three structures appear that can be related to the ionization from the three orbitals $6a_g$, $5b_{1u}$, and $4b_{2u}$ of the neutral molecule (shown in green, red, and purple, respectively). The signal is integrated over 150 meV energy widths, in order to extract the time-constants. **b** Time-dependent electron signal for the three structures discussed in (**a**). Three time-constants are extracted from a simple fitting procedure (see Methods). **c** The different High Harmonic spectra used in the experiment. **d** Extracted lifetimes for the three zones discussed in (**a**), for different high harmonic spectra (the same color code has been used as in (**c**)). **e** (left) The relevant part of the cationic spectrum of Naph produced upon XUV ionization computed with ADC(3) method. The states are populated by ionization from the $6a_g$, $5b_{1u}$, and $4b_{2u}$ orbitals of the neutral molecule (shown in dark green, dark brown, and dark blue, respectively). The intensity of each line reflects the multielectronic character of the state, as well as its initial population. (right) Time-dependent populations obtained via the constructed vibronic-coupling model and integration over the corresponding 150 meV energy windows. **f** Extracted lifetimes for the three zones at 0.4, 0.65, and 0.9 eV (shown in green, red, and purple respectively) as a function of the IR intensity

orbitals) and no ultrafast dynamics. If non-Born−Oppenheimer effects are present (see Fig. 3c), population may be transferred between the electronic states due to the nonadiabatic couplings. In this situation it is expected that higher energy states are depopulated and the population is transferred to lower energy states. In the case considered here, both multielectronic states and non-Born−Oppenheimer effects are present. Due to the multi-electronic effects, when approaching the double-ionization threshold, the density of states may increase dramatically (see Fig. 3e). In principle, when states get closer, the nonadiabatic effects get stronger, suggesting that with the increase of the state energy, the relaxation time should become shorter. The nonadiabatic relaxation should, however, occur through a large number of CIs, making the dynamics of many strongly entangled electrons and nuclei highly nontrivial (Fig. 3f). In the following, we will show that this leads to the specific behavior observed in

our experiment in which the population is trapped for a much longer period of time than expected. We note that in previous experiments, only time-resolved ionic yields could be measured, allowing to trace either simple dynamics or averaged relaxation processes[22,23]. Our present approach gives access to the relaxation of energy-resolved density of states and therefore allows for a much more refined investigation which requires a complex theoretical description that takes into account a large number of strongly coupled states as described below.

**Signature of coupled multielectronic and non-Born−Oppenheimer dynamics**. To simulate the many-body quantum dynamics, we constructed a vibronic-coupling model based on 23 electronic states and 25 active vibrational modes (Methods). The cationic eigenstates have been computed with the non-Dyson

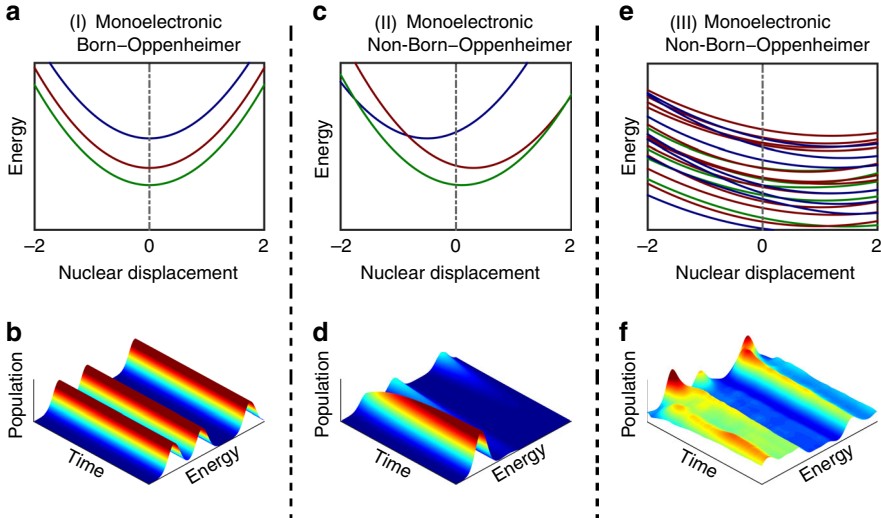

**Fig. 3** Relaxation dynamics of naphthalene at different levels of approximation. Case I where monoelectronic and Born−Oppenheimer (adiabatic) approximations are considered: **a** Potential energy surfaces of uncoupled electronic states and **b** corresponding time-dependent populations, which shows no transfer of population between states. Case II where monoelectronic and non-Born−Oppenheimer (non-adiabatic) approximations are considered: **c** Potential energy surfaces of states with vibronic couplings (conical intersections) and **d** corresponding time-dependent populations which shows that population can be transferred between the states. Case III where both monoelectronic and Born−Oppenheimer approximations break down. This last panel shows the situation studied in the present work. In this regime, due to the multielectronic effects, a quasi-continuum of electronic (shake-up) states is formed and the population between them can be transferred very efficiently due to the strong nonadiabatic effects resulting in a large number of conical intersections. A naive interpretation will lead to the conclusion that the time scale decreases with the increase of the energy in contradiction with experimental observation. **e** Computed potential energy surfaces of naphthalene taking into account for multielectronic and non-Born−Oppenheimer effects and **f** the corresponding time-dependent populations that show that three main contributions appear. Like in the experiment, we observe that slower time scale is associated with a higher energy state

ADC(3) Green's function method[24]. The method takes into account the multielectronic effects up to third order of perturbation theory. All cationic states with a binding energy $E_i$ lying below the XUV photon energy can be populated. However, in the experiment, the weak IR pulse ionizes only states lying just below the double-ionization potential. The cationic eigenstates in this energy range are shown in Fig. 2e. A comparison of the computed cationic spectrum and the experimental TR-PES map shows that the three features observed experimentally around 0.4, 0.65, and 0.9 eV can be directly related to three groups of states that stem from the ionization out of three inner-valence orbitals: $6a_g$, $5b_{1u}$, and $4b_{2u}$. The measured time-constants correspond, therefore, to the timescale of the nonadiabatic relaxation dynamics of these groups of states. This dynamics was obtained by simultaneously propagating the nuclear wavepackets on all nonadiabatically coupled multielectronic states using the MCDTH method[25,26], taking into account the initial population of the states upon the XUV ionization.

Looking only at the most populated states within each of the three groups, computed to be at $E_i = 19.54$, 19.74, and 20.10 eV (Fig. 2e), we obtain relaxation times of 8.1, 8.2, and 5.5 fs, respectively. These time-constants not only differ significantly from the experimentally observed ones, but show also a different trend: lower in energy states relax slower, as expected for the monoelectronic/non-Born−Oppenheimer case (see Fig. 3d). In fact, although the respective populations leak out very fast from these three states, they are transferred to very close-lying states, which cannot be resolved, but where those populations can be trapped for longer. We can account for this by integrating the population evolution of all symmetry-related shake-up states lying within 150 meV below the three most populated ones, yielding the following time-constants: 15 fs

around 0.4 eV, 21 fs around 0.64 eV, and 35 fs around 0.88 eV (Fig. 2c).

These results are not only much closer but also show the trend observed experimentally. As a result, the physical picture seems to be clear. The correlation effects become stronger going deeper in the valence shell. Therefore, removing a deeper electron results in populating an increasing number of states. This leads to a complex shake-up zone with a large number of CIs. Due to the high density of electronic states and because of the ultrashort lifetime of these states, the individual electronic states cease to be an adequate description of the transient molecular state. The number of CIs through which the wavepacket has to go increases with the binding energy which leads to an increase of the relaxation time as observed in the experiment. This suggests that the observed dynamics is intrinsic to the strongly nonadiabatically coupled multielectronic states. We note that increasing the number of states considered in the model will lead to even more accurate description of the many-body molecular wavefunction and will bring the relaxation times even closer to the experimental values. A better description of the shake-up zone can be done, for example, by increasing the basis sets used in the ADC calculations, which however will face a prohibitive computational time.

**XUV-induced vibronic coherences.** Let us now turn to our second important observation. We note that the calculations predict smooth oscillations in the population decay around 100 fs, which is illustrated in Fig. 4a with the time-dependent population of the main state mapped to 0.4 eV. Comparing to the experimental data, and taking into account the cross-correlation between the XUV and IR pulses, we see a similar recurrent signal (Fig. 4b, c). The

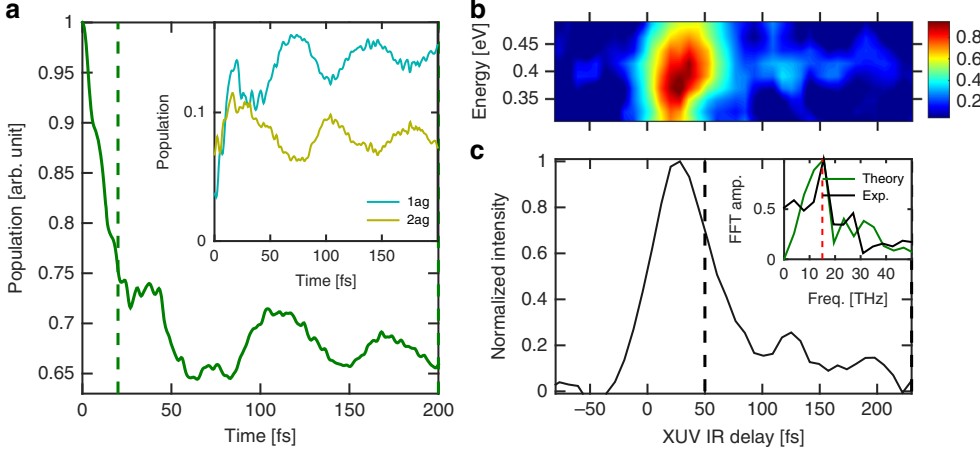

**Fig. 4** Long-lived vibronic coherence. **a** Computed time-dependent population following ionization from $6a_g$ orbital (see Fig. 2c) computed with the help of the constructed vibronic-coupling model. The periodic structures are attributed to the $a_g$ vibrational modes (see inset). The $2a_g$ state belongs to the experimental probing region whereas $1a_g$ lies below. **b** Measured time-dependent electron spectrum mapping the relaxation of the $6a_g$ states. **c** Experimental signal integrated over the electron kinetic energy in the range $0.4 \pm 0.075$ eV. The inset shows the FFT of the experimental signal taken from 50 to 250 fs (black) compared with the theory taken from 20 to 200 fs (green). This leads to an oscillation period of 65 fs

analysis shows that these recurrences correspond to coherent vibrational dynamics of the low-frequency $a_g$ mode at 514.3 cm$^{-1}$ (inset of Fig. 4a) that couples two electronic states, one lying within ($2a_g$) and one lying below ($1a_g$) the energy region probed experimentally, making the wavepacket experimentally accessible only during the time it evolves on $2a_g$ surface. This mode is of particular importance, as it is the slowest totally symmetric mode of the molecule and thus easy to excite. Moreover, many states in this energy region share the same symmetry and are therefore coupled by this particular mode. We conclude that the observed dynamics correspond to cationic excited molecules that coherently vibrate. We note that the effect is observed only at the lowest electron energy (0.4 eV), which corresponds to the ionization out of $6a_g$ orbital. Neither the calculations, nor the experiment show oscillations in the dynamics triggered by the ionization of the other two orbitals $5b_{1u}$ (mapped to 0.64 eV signal) and $4b_{2u}$ (mapped to 0.88 eV). We also note that this effect is not observed in the case of Ada, which we attribute to the fact that slow vibrational mode are not active for Ada due to its compact structure.

While XUV-induced coherent dynamics have been observed in small molecules[27], its existence in large polyatomic systems like Naph has never been proven to date. Light-induced vibrational coherences is known for low excited molecular states[28]; it is striking that such coherence can also survive the molecular complexity in highly excited states.

In our experiment, relaxation dynamics with timescales that increase with the state energy has been observed in XUV-excited Naph and Ada molecules. This behavior is intrinsic to the many-body quantum nature of the molecule, i.e. their fully entangled multielectronic and non-Born−Oppenheimer character. These results open new avenues in controlling molecules using the entangled quantum properties of their constituting particles. We have also observed XUV-induced vibronic coherence in Naph. The observation of such quantum effect at low excitation energy has led to suggesting that quantum coherence might play an important role in life-related phenomena such as photosynthesis[29–31]. Similarly, our observation raises the question of the role of the ultrafast quantum many-body dynamics initiated by the illumination of molecules with energetic radiation in space[32]

which predetermines the chemical reactivity of the irradiated species.

## Methods

**Experimental setup**. The setup (Supplementary Fig. 1) consists of a Mach−Zender interferometer coupled with a VMI spectrometer. The output of a commercial amplified laser system producing pulses of 2 mJ, centered around 810 nm, 5 kHz repetition rate and 25 fs pulse duration, is split into two arms. One part is focused, with a 30 cm lens, inside a windowless gas cell filled with rare gas, 2 mm thickness, where the fundamental frequency is up-converted by means of high-order harmonic generation (HHG). The pressure inside the gas cell is maintained around 10 mbars. The XUV frequency comb generated under such conditions is composed of odd harmonics from the 7th (H7) up to the 21st (H21). The HH spectrum can be tuned by adjusting the phase-matching conditions and by choosing an adequate rare gas (xenon, krypton or argon). The contribution of the XUV radiation that is due to the long electron trajectories in the HHG process is filtered out using a motorized iris just after the cell and then the copropagating IR is removed in two stages. First, a beam splitter with $Nb_2O_5$ outermost layer reflects mainly the XUV part. Second, the XUV beam passes through a 200 nm Al filter that transmits light only above 17 eV. The XUV beam is focused by a 30 cm toroidal mirror. A gold-coated grating mounted on a rotational stage can be inserted to measure the XUV spectra at the first order of diffraction. Typical spectra optimized to variable order (around H13 up to H25 is shown in Supplementary Fig. 2). The XUV beam then passes through a 45° drilled mirror (3 mm hole diameter), and the other arm of the interferometer is time delayed and recombined with the XUV beam by a reflection at the vicinity of the hole. Both beams are focused in the interaction region of the VMI spectrometer. The energy of the second arm of the interferometer is controlled by a half–wave plate combined with a thin film polarizer (vertical polarization). Then, the beam is reflected by flat mirrors. Each mirror is placed on a manual translational stage, in order to reach an approximate temporal overlap between the pump and the probe pulses. The beam then passes through a refractive delay line, consisting of a pair of parallel wedges, which allows to introduce a time delay proportional to the displacement of the wedges by a linear actuator. Finally, the beam passes through a 75 cm lens and is reflected at the vicinity of the hole of the 45° drilled mirror.

The VMI spectrometer follows the standard design introduced by Eppink et al.[33]. Electron trajectories are focused using a static electrostatic lens and detected by a double Micro-Channel Plate (MCP) and a phosphor screen assembly imaged on a CCD camera. The two molecules studied in our experiment, Naphthalene and Adamantane (Supplementary Fig. 3), have a sufficient partial pressure at room temperature to be directly injected into the vacuum chamber through a capillary. The capillary is connected to the repeller electrode of the VMI and the molecules are ejected towards the detector and cross the XUV-IR beams.

**Data analysis**. The VMI images are Abel inverted and angularly integrated to obtain the kinetic energy spectrum. Time-resolved electron kinetic energy spectrum has been recorded and the two-color signal has been extracted by subtracting

the contributions from the XUV and NIR pulse at each delay. The time-dependent signal obtained at each electron kinetic energy is integrated over a $\Delta E = 150$ meV energy range, leading to a spectrum that is used to extract the decay times as described in the following.

The measured signal can be schematically described as follows: excited states are produced at $t = t_0$, part of the initial population relaxes following an exponential decay $\tau_{decay}$ while a fraction of the initial population remains constant. The temporal resolution is determined by the cross-correlation between the XUV and IR pulses. This cross-correlation of duration $\tau_{crossco}$ is represented by a Gaussian function. Therefore, we use the following formula to extract the decay lifetime from the electron signal obtained at a given kinetic energy:

$$\Delta S_{Fit} = \exp\left(-4\ln(2)\left(\frac{t}{\tau_{crossco}}\right)^2\right)$$
$$\otimes \left[\theta(t-t_0)\left(A_{decay}\exp\left(\frac{-(t-t_0)}{\tau_{decay}}\right) + A_{step}\right)\right], \quad (1)$$

where $\tau_{crossco}$ and $t_0$ are fixed parameters (that can be extracted from the high $KE_e$-features) and $A_{decay}$, $\tau_{decay}$, and $A_{step}$ are free parameters. $\theta(t - t_0)$ is the Heaviside step function.

**Experimental results for naphthalene**. The photoelectron signal arises from the photoionization of the XUV excited cations. This assumption is also validated by the comparison between the time-dependent dication signal obtained by measuring the variation of the ion yield, and the time-dependent low-energy electron signal that we integrate over energy. In the first case, the fitting procedure leads to a time scale of $36 \pm 4$ fs and in the second case we obtain $40 \pm 7$ fs (Supplementary Fig. 4).

The error bar on the extracted decay time-constant is defined as the error to the fit. We have also measured decay time constant defined as the central value of a statistical set of independent measurements. The associated error bar is then given by the dispersion of the experimental results. We have included in the statistics the measurements performed at different probe intensities and with different XUV spectra, since no significant changes were observed in the dynamics while changing such experimental parameters. These results are presented in Supplementary Fig. 5.

**Comparison between naphthalene and adamantane**. Similar measurements have been performed for adamantane (Supplementary Fig. 3b). The results of the measurement are presented in Supplementary Fig. 6. They show a very similar trend as observed in the case of naphthalene: the extracted time scale increases with the electron kinetic energy from $25 \pm 3$ to $80 \pm 20$ fs.

**Theory**. The ionization spectra of the naphthalene molecule have been calculated with the third order non-Dyson Algebraic Diagrammatic Construction scheme (nd-ADC(3))[34,35] for representing the one-particle Green's function. The spectrum at the ground state equilibrium geometry is shown in Supplementary Fig. 7. Each line in the spectrum corresponds to a cationic eigenstate of the molecule. The intensity of each line is given by the weight of the one hole (1h) contributions to the corresponding cationic state. The missing to one part of the line thus reflects the multielectronic contribution to the corresponding state, which at third order means all two-hole-one-particle (2h1p) contributions. In other words, even though only the 1h part of the cationic state is detected experimentally, its multielectronic nature is encoded through its initial population (i.e. the intensity of the line) and its energy position. We see that within the energy range accessible in the present experiment (close to the double-ionization threshold (DIT), between approximately 20 and 19 eV), the majority of the states come from the ionization out of only three orbitals. These orbitals are $6a_g$, $5b_{1u}$, and $4b_{2u}$. The fact that only three symmetries out of eight (the naphthalene molecule belongs to $D_{2h}$ symmetry point group) are present will limit the number of relevant vibrational modes. The symmetry selection rules[36] reduce the problem to the following coupling scheme: Apart from the totally symmetric $a_g$ mode, the states of interest can be coupled by the nonsymmetric $b_{1u}$ mode (coupling the states of $a_g$ and $b_{1u}$ symmetry), $b_{2u}$ mode (coupling $a_g$ and $b_{2u}$) and $b_{3g}$ mode (coupling $b_{1u}$ and $b_{2u}$). The vibrational modes have been obtained at the MP2 level using the GAUSSIAN program package[37]. The ionization spectra of the naphthalene have been computed along all the vibrational modes that belong to the four symmetries $a_g$, $b_{1u}$, $b_{2u}$, and $b_{3g}$, up to 2 $Q_i$ with step of 0.25 $Q_i$, where $Q_i$ is the dimensionless coordinate associated with the normal mode $i$. All calculations have been performed using cc-pVDZ basis set[38].

**Vibronic-coupling Hamiltonian**. In order to simulate the coupled nuclear and electronic dynamics triggered by the inner-valence ionization of naphthalene, we have built a vibronic-coupling Hamiltonian[39] for 23 electronic states present in the energy range of interest, and 25 relevant vibrational modes. The included states, 5 states of $a_g$ symmetry, 11 of $b_{1u}$ symmetry, and 7 of $b_{2u}$ symmetry, are listed in Supplementary Table 1 and consist of all ionic states in the range 19.02–20.21 eV, which can be unambiguously identified and followed along the whole distortion range of the vibrational modes. The vibrational modes that can couple the states of

interest belong to $a_g$, $b_{1u}$, $b_{2u}$, and $b_{3g}$ irreducible representations and are listed in Supplementary Table 2. The vibronic-coupling Hamiltonian can be written as:

$$H = \tau_N + \nu_0 + W \quad (2)$$

where $\tau_N$ and $\nu_0$ denote the kinetic and the potential energy of the neutral unperturbed reference ground state, respectively. Using a harmonic approximation for the vibrational modes, $\tau_N$ and $\nu_0$ can be written as:

$$\tau_N = -\frac{1}{2}\sum_i \omega_i \frac{\partial^2}{\partial Q_i^2}, \quad (3)$$

$$\nu_0 = \frac{1}{2}\sum_i \omega_i Q_i^2 \quad (4)$$

with $\omega_i$ being the frequency of mode $i$. The matrix $W$ in Eq. (2) contains the diabatic cationic states, and the couplings between them. Using the standard for the vibronic-coupling theory Taylor expansion of the matrix elements, $W$ can be written as:

$$W_{jj} = E_j + \sum_{i\in a_g}\kappa_i^j Q_i + \sum_{i\in a_g,b_{1u},b_{2u}}\gamma_i^j Q_i^2 \text{ for } j\in a_g, \quad (5)$$

$$W_{jj} = E_j + \sum_{i\in a_g}\kappa_i^j Q_i + \sum_{i\in a_g,b_{1u},b_{3g}}\gamma_i^j Q_i^2 \text{ for } j\in b_{1u}, \quad (6)$$

$$W_{jj} = E_j + \sum_{i\in a_g}\kappa_i^j Q_i + \sum_{i\in a_g,b_{2u},b_{3g}}\gamma_i^j Q_i^2 \text{ for } j\in b_{2u}, \quad (7)$$

$$W_{jk} = \sum_{i\in a_g}\lambda_i^{j,k} Q_i \text{ for } j,k\in a_g; j<k, \quad (8)$$

$$W_{jk} = \sum_{i\in a_g}\lambda_i^{j,k} Q_i \text{ for } j,k\in b_{1u}; j<k, \quad (9)$$

$$W_{jk} = \sum_{i\in a_g}\lambda_i^{j,k} Q_i \text{ for } j,k\in b_{2u}; j<k, \quad (10)$$

$$W_{jk} = \sum_{i\in b_{1u}}\lambda_i^{j,k} Q_i \text{ for } j,k\in a_g, b_{1u}\cup b_{1u}, a_g; j<k, \quad (11)$$

$$W_{jk} = \sum_{i\in b_{2u}}\lambda_i^{j,k} Q_i \text{ for } j,k\in a_g, b_{2u}\cup b_{2u}, a_g; j<k, \quad (12)$$

$$W_{jk} = \sum_{i\in b_{3g}}\lambda_i^{j,k} Q_i \text{ for } j,k\in b_{1u}, b_{2u}\cup b_{2u}, b_{1u}; j<k. \quad (13)$$

In the above expressions, $E_j$ is the vertical ionization energy of state $j$, $\kappa_i^j$ and $\gamma_i^j$ are the linear and quadratic coupling parameters of state $j$ for normal mode $i$, respectively, and $\lambda_i^{jk}$ is the linear coupling parameter between states $j$ and $k$ by the normal mode $i$. These quantities are obtained directly from the ab initio nd-ADC(3) calculations through a least-square fit procedure performed by the VCHam module, included in the Heidelberg Multi-Configuration Time-Dependent Hartree (MCTDH) program package (http://www.pci.uni-heidelberg.de/tc/usr/mctdh/doc/). An example of the result of the fitting procedure is shown in Figure S8. Our analysis shows that the high-frequency CH vibrational modes lead to a very weak coupling between states. That is why we have neglected the 8 modes of this type, leading to a vibronic-coupling Hamiltonian based on 25 modes (presented in Supplementary Table 2).

**MCTDH calculations**. The vibronic-coupling Hamiltonian was used to propagate nuclear wavepackets on the coupled manifold of electronic states via the MCTDH methods[40]. MCTDH is a powerful grid-based method for numerical integration of the time-dependent Schrödinger equation, particularly suitable for treating multidimensional problems[41]. The Heidelberg MCTDH package has been used for this calculation.

We have performed three independent propagations describing the dynamics triggered by ionization out of $6a_g$, $5b_{1u}$, and $4b_{2u}$ orbitals. The initial population has been distributed between 5, 11, and 7 states, respectively, according to the spectral intensity of the corresponding ionic state at equilibrium geometry (Supplementary Fig. 7). These wavepackets have been then propagated for 200 fs taking into

account all coupled 23 states. The results are presented in Supplementary Fig. 8. On the left set of panels, we report the evolution of the populations of the states located within an energy window of 150 meV below the initially most populated state (i.e., having the largest intensity at equilibrium geometry) in each group. The energy window chosen reflects the experimental resolution. We see that although the initial population is transferred very fast to lower lying states (time-constants of ~ 8.1, 8.2, and 5.5 fs are obtained for the relaxation of $a_g$, $b_{1u}$, and $b_{2u}$ states, respectively), part of the population is transferred and somewhat trapped in close-lying states. As the latter states lie within the experimental resolution, for simulating the experimental observation we need to sum all populations that lie within the energy-resolution window. The result of this procedure is shown in the right set of panels of Supplementary Fig. 8.

We thus obtain the following relaxation timescales: 15 fs for the $a_g$ states, 21 fs for the $b_{1u}$ states, and 35 fs for the $b_{2u}$ states. Although the computed time-constants are about 30% smaller than those extracted from the experiment, they correctly reproduce the general trend, namely the states closer to the DIT relax slower. The explanation for this somewhat counter-intuitive result can be well understood with the increasing density of states when approaching the DIT. In dense spectral regions, the wavepackets have to go through a larger number of CIs in their relaxation path, compared to more sparse spectral regions, thus resulting in a slower relaxation time. This also explains the shorter relaxation times obtained in our theoretical modeling. Although quite advanced, our model only includes a fraction of all the states in the energy range of interest.

**Code availability**. The ADC and MCTDH custom codes are available upon request with no restrictions.

## Data availability

All other relevant data supporting the key findings of this study are available within the article and its Supplementary Information files or from the corresponding authors upon request.

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

## Acknowledgements

The research has been supported by CNRS, ANR-16-CE30-0012 "Circé" programme Blanc, Fédération de physique Marie-Ampère, Région Aquitaine (Caracatto 20131603008). V.D. acknowledges the financial support of DFG through QUTIF priority programme. A.I.K. thanks US ARO for financial support under grant No W911NF-14-1-0383. We thank Horst Köppel for fruitful discussions.

## Author contributions

F.L. and A.I.K. led and designed the research program and wrote the manuscript. A.I.K. and V.D developed and performed the calculations. A.M., V.L., G.K., and M.H performed the measurements and analyzed the experimental data. E.C. and L.Q developed an early version of the XUV beamline. F.C. and C.J. have contributed to the discussion of the results. All authors have contributed to the final version of the manuscript.

**Additional information**

**Competing interests:** The authors declare no competing interests.

