## [Peer Review File · Nature Communications]

Reviewers' comments:

Reviewer #1 (Remarks to the Author):

This paper by A. Marciniak et al. presents the result of a combined experimental-theoretical investigation of the effect of strongly non-adiabatically coupled multielectronic states on the photoelectron dynamics in a polyatomic molecule upon single photon ionization in the XUV range between 17 and 25 eV.

Although the combination of a VMI spectrometer and a pump XUV – IR probe time resolved technique is a well established technique (see for example Remetter et al, Nature Physics 2, 323–326, 2006), the presented experimental results and their interpretation supported by the theory are noteworthy. However I cannot recommend the paper to be published in Nature Communication in the present form with the following reasons:

1. In the first lines of the introduction I would update the cited literature with the recent works on molecular dynamics from the Hans Jacob Wörner and Ursula Keller groups together with S. Haessler et al. Phys. Rev. A 80, 011404 (2009). Moreover, still in the introduction, I would mention the theoretical work from M. Vacher et al. Phys. Rev. Lett. 118, 083001 (2017), dedicated exactly to polyatomic molecules and nuclear-electronic coupled dynamics.
2. There is a discrepancy between the XUV pulse train bandwidth claimed in the main text: 17-35 eV and what is shown in the supplementary information: figure S2. I think in the main text the intention was to write 17-25 eV. Are the spectra in S2 acquired without the Al filter? Because I can see harmonics below 15 eV which is the Al threshold. Moreover I would add a sentence clarifying why, even though the XUV photon energies (17-25 eV) exceed the IP2 threshold (21.5 eV), you do not observe double ionization from single photon absorption, and in case you do, why you can neglect it.
3. In the sentence starting with: "the time dependent two color...respectively." I would write $E_{kin} \sim 0.4$ to be consistent with the rest of the manuscript, or better 0.4 ± 0.75 eV; 0.65 ± 0.75 eV and 0.9 ± 0.75 eV. If the integration intervals are not symmetric, as it appears in Fig. 2a, I would motivate the specific choice.
4. Figure 1 is not sufficiently accurate. I would at least put the colorbar in the VMI picture, the direction of the visualized momenta and the quantitative energy axis in 1b. Figure 1c is not even described through the entire text!
5. In general the captions of the figures are not nicely written and should be completely revised:
 - a. Color consistency in Fig.2a and 2b: there is no green, brown and blue!

- b. In caption 2b) I assumed you intended to write “discussed in a)”.
 - c. Missing legends Figure 3a, c, and e defining the three main contributions.
 - d. The caption in figure 4 refers to a missing Fig. 1d and in caption c) is not specified the energy range of the integrated photoelectron spectrum.
6. I found not easy to read the explanation how the theory could reproduce the experimental trend and the time decay values. In particular I refer to the paragraph:” Looking only...0.9 eV (see Figure 2.c).” I would clearly write what each of the conditions (I, II, III) represented in Fig. 2 can resolve and what not, perhaps mentioning about the shake up processes already 2 paragraphs before, i.e.: “ In the case considered herenon-trivial (fig.3f)”.
7. I would include in the main text, and not only in the Methods/Theory, why the calculated decay times deviated of about 10 fs from the experimental.
8. To my knowledge, indeed I cannot find any other works presenting XUV induced coherent dynamics in polyatomic molecules like Naphthalene. This is extremely interesting but the comparison with the theoretical calculations is poorly presented in only a couple of sentences. A comparison in the frequency domain between theory and experiment could already help. Why did you choose only the lowest range at 0.4 eV? In Fig.2a, in fact, I can see a close to periodic signal also at 0.9 eV. Can you comment on that?

Reviewer #2 (Remarks to the Author):

The manuscript entitled "Dynamics of Many-body Quantum Molecular States Revealed by Ultrashort XUV pulses" by Dr Lepine and colleagues, represents an interesting experimental-theoretical work, on a topic that is of high relevance in the field.

The many-body interactions have been implicated in new mechanisms for motion of charge across the molecules on few-femtosecond and attosecond timescales. However, the experimental results on this question have been difficult to obtain due to the complexity of the problem. The work here indicates that a one-to-one comparison between experimental and theoretical results could help to illustrate the role of electronic correlations and electron-nuclear couplings.

The paper is quite interesting, and the results seem solid, however, the presentation is a bit abrupt in the main manuscript, and lacking in some important details. Below are some specific questions:

1. The paper main focus is on the relaxation dynamics of multi-electron states in naphthalene and Adamantane. The relaxation timescales are inferred from decaying second (IR) ionization signal, but authors do not discuss the actual relaxation pathways and products. Does the decay simply refer to the downward transitions in the Franck-Condon region to some final level? Do these excited cations fragment? If so, how does the fragmentation impact the photoelectron yield?

2. Authors report the delay dependence of photoelectron yields in figure 2 as their main set of results, but do not provide the XUV harmonic spectrum used to obtain that data, except the fact that it extends from 17-35 eV. As harmonic 11 and 13 are below IP2 by 4.5 eV and 1.35 eV respectively, they cannot contribute to the 2 color electron yields reported in figure 2 at electron energies 0.35, 0.65, and 0.9 eV. The next harmonic (15th) is above the IP2 by 1.75 eV, and from figure S5 it seems to have a bandwidth of 0.3 eV. The IR dressing of direct photoelectrons can thus result in signals at $(1.75-1.55) \pm 0.15$ eV, i.e. between 0.05-0.35 eV. This contribution can enhance the 0.35 eV electron signals levels around time zero, which may manifest as a faster apparent decay. This process should not affect the analysis of photoelectrons at 0.65 and 0.9 eV. How do authors take into account of such IR dressing contributions?

3. Other related experimental questions are: (a) Was the IR polarization parallel to the XUV? Did authors study the dependence of two-color electron yield on IR polarization, or the angular distribution photoelectrons? (b) Authors mentioned they varied the intensity of IR probe. How does that effect the 2 color photoelectron signals? Did they discount the possibility of 2 IR photon transitions? (c) Are there any neutral Rydberg states below IP2 that could be excited by the XUV?

4. In general, the observation of oscillations corresponding to vibrational coherences in figure 4 requires that the wave packet be accessible at specific coordinates or times, but not at others. What provides this selectivity in the experiment? Also from the inset, it seems that 1ag and 2ag are out-of-phase. In such a case, what determines the observed oscillation contrast?

5. The IR pulse duration is listed as 25 fs, however, the XUV pulse duration is not explicitly mentioned. In reference to equation 1 of the supplement, I am also curious as to what is the value of τ_{crossco} extracted from higher energy features ?

Overall, taken together with extensive supplementary information, the paper represents detailed effort on gaining new understanding of multi-electronic and non-BO effects in molecules. The experimental and theoretical results compliment each other. I would recommend publication the

manuscript provided the above questions can be addressed and manuscript can be improved by transferring few pieces of vital information from the supplement to the manuscript.

Reviewers' comments:

Reviewer#1(Remarks to the Author):

1. In the first lines of the introduction I would update the cited literature with the recent works on molecular dynamics from the Hans Jacob Wörner and Ursula Keller groups together with S. Haessler et al. Phys. Rev. A 80, 011404 (2009). Moreover, still in the introduction, I would mention the theoretical work from M. Vacher et al. Phys. Rev. Lett. 118, 083001 (2017), dedicated exactly to polyatomic molecules and nuclear-electronic coupled dynamics.

Indeed, the text was missing some important recent results in relation to attosecond molecular science. We have now added the following references:

J. Vos, L. Cattaneo, S. Patchkovskii, T. Zimmermann, C. Cirelli, M. Lucchini, A. Kheifets, A. S. Landsman, U. Keller “Orientation-dependent stereo Wigner time delay and electron localization in a small molecule” Science 360, 6395, 1326-1330 (2018).

S. Haessler, B. Fabre, J. Higuët, J. Caillat, T. Ruchon, P. Breger, B. Carré, E. Constant, A. Maquet, E. Mével, P. Salières, R. Taïeb, and Y. Mairesse Phase-resolved attosecond near-threshold photoionization of molecular nitrogen Phys. Rev. A 80, 011404, (2009).

Y. Pertot, C. Schmidt, M. Matthews, A. Chauvet, M. Huppert, V. Svoboda, A. von Conta, A. Tehlar, D. Baykusheva, J.-P. Wolf, H. J. Wörner, Time-resolved x-ray absorption spectroscopy with a water window high-harmonic source, Science 355(6322), 264-267 (2017).

L. Cattaneo, J. Vos, R. Y. Bello, A. Palacios, S. Heuser, L. Pedrelli, M. Lucchini, C. Cirelli, F. Martín, U. Keller, Attosecond coupled electron and nuclear dynamics in dissociative ionization of H₂, *Nature Physics*, 14, 733-738 (2018).

M. Vacher, M. J. Bearpark, M. A. Robb, and J. P. Malhado Electron Dynamics upon Ionization of Polyatomic Molecules: Coupling to Quantum Nuclear Motion and Decoherence *Phys. Rev. Lett.* 118, 083001 (2017).

2. There is a discrepancy between the XUV pulse train bandwidth claimed in the main text: 17-35 eV and what is shown in the supplementary information: figure S2. I think in the main text the intention was to write 17-25 eV.

The high harmonic spectra presented in the S.I. were plot in High Harmonic order, which indeed was confusing with the rest of the text where we used “eV”. We have now plotted all the spectra using “eV” as units.

The XUV spectrum used in the experiment was typically within 17 and 35 eV (although the highest harmonics have low intensities). To clarify this point, we have added the following sentence in the main text:

“The presented experimental results have been obtained with a HH spectrum containing harmonics between 17 and 35 eV, centered around 26 eV”

Are the spectra in S2 acquired without the Al filter? Because I can see harmonics below 15 eV which is the Al threshold. Moreover

All the spectra were acquired with an Al filter. Indeed, no harmonics are observed in these spectra below 15 eV (HH11). This is now specified in the figure caption of supplementary Figure 2:

“These spectra have been acquired using an Al filter with a transmission threshold that allows for transmitting only harmonics above 15eV.”

I would add a sentence clarifying why, even though the XUV photon energies (17-25 eV) exceed the IP2 threshold (21.5 eV), you do not observe double ionization from single photon absorption, and in case you do, why you can neglect it.

The referee is correct, the high harmonics above the second IP indeed produce double ionization (dications produced from the neutrals). However, only a small amount of dication is observed in the mass spectrum (as the ion yield measured with a time-of-flight mass spectrometer shows). More importantly, in our analysis we consider only the “two-color signal”, meaning that the electrons produced directly from the XUV excitation (“XUV only” signal) have been subtracted and we consider only electrons produced by both XUV and IR fields. In order to clarify this point, we have added the following sentence:

“by doing so, we subtract the contribution of electrons originating from single and double ionization of the molecule by one XUV photon.”

3. In the sentence starting with:“ the time dependent two color...respectively.” I would write $E_{kin} \sim 0.4$ to be consistent with the rest of the manuscript, or better 0.4 ± 0.75 eV; 0.65 ± 0.75 eV and 0.9 ± 0.75 eV. If the integration intervals are not symmetric, as it appears in Fig. 2a, I would motivate the specific choice.

The interval is indeed symmetric but the central energy was not accurately mentioned. We have changed the text and indicated the accurate electron kinetic energy values and specified that we considered the maxima in the electron kinetic energy spectrum:

“The time-dependent two-color TR-PES is plotted in Fig 2.a. In this map we distinguish 3 features, corresponding to 3 maxima in the spectrum: around 0.4 eV, 0.64 eV and 0.88 eV, respectively. By fitting the signal at these energies integrated between ± 0.75 eV...”

4. Figure 1 is not sufficiently accurate. I would at least put the colorbar in the VMI picture, the direction of the visualized momenta and the quantitative energy axis in 1b. Figure 1c is not even described through the entire text!

We have modified figure 1, its caption and modified the following sentences in the text:

“To demonstrate this, we have investigated carbon-based molecules Naphthalene (Naph, C₁₀H₈) and Adamantane (Ada, C₁₀H₁₆) using an XUV-pump–IR-probe scheme as schematically presented in Fig. 1.a. »

“The XUV pulse populates excited cationic states, with energies just below the double-ionization threshold of the molecule (IP₂ = 21.5 eV) (Fig. 1.b).”

“Hence, the time-resolved photoelectron spectrum (TR-PES) probed the dynamics of the many-body quantum cationic states produced upon XUV photoionization (Fig 1.c) »

5. In general the captions of the figures are not nicely written and should be completely revised:

a. Color consistency in Fig.2a and 2b: there is no green, brown and blue!

We intentionally used different colors for experiment and theory to not confuse the reader.

However, there was indeed a mistake in the reference to the colors in the figure caption. This has now been corrected:

“Three structures appear that can be related to the ionization out of the three orbitals 6ag, 5b1u and 4b2u of the neutral molecule (shown in green, red, and purple, respectively) »

b. In caption 2b) I assumed you intended to write “discussed in a)”.

The referee is correct and this has been corrected.

c. Missing legends Figure 3a, c, and e defining the three main contributions.

We have completely changed the caption as follows in order to make it more explicit. We have changed the figure accordingly:

(I) case where monoelectronic and Born-Oppenheimer (adiabatic) approximations are considered, (a) Potential energy surfaces of uncoupled electronic states and (b) corresponding time-dependent populations, which shows no transfer of population between states.

(II) case where monoelectronic and non-Born-Oppenheimer (non-adiabatic) approximations are considered (c) Potential energy surfaces of states with vibronic couplings (conical intersections) and (d) corresponding time-dependent populations which shows that population can be transferred between the states.

(III) case where both monoelectronic and Born-Oppenheimer approximations break down. This last panel shows the situation studied in the present work. In this regime, due to the multielectronic effects a quasi-continuum of electronic (shake-up) states is formed and the population between them can be transferred very efficiently due to the strong non-adiabatic effects resulting in a large number of conical intersections. A naive interpretation will lead to the conclusion that the time scale decrease with the increase of the energy in contradiction with

experimental observation. (e) Computed potential energy surfaces of naphthalene taking into account for multielectronic and non-born-Oppenheimer effects and (f) the corresponding time-dependent populations that show that 3 main contributions appear. Like in the experiment, we observe that slower time scale is associated to higher energy state .”

d. The caption in figure 4 refers to a missing Fig. 1d and in caption c) is not specified the energy range of the integrated photoelectron spectrum.

We have now changed the figure caption to include the missing information.

“ a) Computed time-dependent population following ionization out of 6ag orbital (see Fig. 2c) computed with the help of the constructed vibronic-coupling model. The periodic structures are attributed to the ag vibrational modes (see inset). 2ag state belong to the experimental probing region whereas 1ag lie below.

b) Measured time-dependent electron spectrum mapping the relaxation of for the 6ag states.

c) Experimental signal integrated over the electron kinetic energy in the range of 0.4 ± 0.75 eV. The inset shows the FFT of the experimental signal taken from 50 to 250 fs (black) compared to the theory taken from 20 to 200 fs (green). This leads to an oscillation period of 65 fs.»

6. I found not easy to read the explanation how the theory could reproduce the experimental trend and the time decay values. In particular I refer to the paragraph:” Looking only...0.9 eV (see Figure 2.c).” I would clearly write what each of the conditions (I, II, III) represented in Fig. 2 can resolve and what not, perhaps mentioning about the shake up processes already 2 paragraphs before, i.e.: ” In the case considered herenon-trivial (fig.3f)”.

We have modified the figure caption of figure 3 (see answer to point 5.) that contains the description of what the 3 situations will lead to.

We have also modified the text as proposed by the referee:

“As shown in Fig 3.a,b, within the mono-electronic and Born-Oppenheimer approximation, one would expect three completely decoupled cationic states (corresponding to the ionization out of three inner-valence orbitals) and no ultrafast dynamics. If non-Born-Oppenheimer effects are present (see Fig 3.c), population may be transferred between the electronic states due to the non-adiabatic couplings. In this situation it is expected that higher energy states are depopulated and the population is transferred to lower energy states. In the case considered here, both multi-electronic states and non-Born-Oppenheimer effects are present. Due to the multi-electronic effects, when approaching the double-ionization threshold the density of states may increase dramatically (see Fig 3.e). In principle, when states get closer, the non-adiabatic effects get stronger, suggesting that with the increase of the state energy, the relaxation time should become shorter. The non-adiabatic relaxation should, however, occur through a large number of CIs, making the dynamics of many strongly entangled electrons and nuclei highly non-trivial (Fig 3.f). In the following, we will show that this leads to the specific behavior observed in our experiment in which the population is trapped for a much longer period of time than expected.»

7. I would include in the main text, and not only in the Methods/Theory, why the calculated decay times deviated of about 10 fs from the experimental.

We have added in the main text the following sentence:

“This suggests that the observed dynamics is intrinsic to the strongly non-adiabatically coupled multielectronic states. We note that increasing the number of states considered in the model will lead to even more accurate description of the many-body molecular wavefunction and will bring

the relaxation times even closer to the experimental values. A better description of the shake-up zone can be done, for example, by increasing the basis sets used in the ADC calculations, which however will face a prohibitive computational time. »

8. To my knowledge, indeed I cannot find any other works presenting XUV induced coherent dynamics in polyatomic molecules like Naphthalene. This is extremely interesting but the comparison with the theoretical calculations is poorly presented in only a couple of sentences. A comparison in the frequency domain between theory and experiment could already help. Why did you choose only the lowest range at 0.4 eV? In Fig.2a, in fact, I can see a close to periodic signal also at 0.9 eV. Can you comment on that?

We have added now in Fig. 4 a comparison between the Fourier transform of the theoretical and experimental signal.

We have shown only the oscillations observed at 0.4 eV because the effect was not clearly observed at higher photoelectron energy (not observed for all laser conditions) while it was robust at 0,4 eV.

This is also in line with theoretical calculations that show oscillations corresponding to ag modes between states lying within and outside the experimental probing region. This is not the case for the two other energy region.

We have added the following text:

“Comparing to the experimental data, and taking into account the cross-correlation between the XUV and IR pulses, we see a similar recurrent signal (see Fig 4.b-c). The analysis shows that these recurrences correspond to coherent vibrational dynamics of the low-frequency ag mode at 514.3 cm⁻¹ (see inset of Fig 4.a) that couples two electronic states, one lying within (2ag) and one lying below (1ag) the energy region probed experimentally, making the wave packet experimentally accessible only during the time it evolves on 2ag surface. This mode is of particular importance, as it is the slowest totally symmetric mode of the molecule and thus easy to excite. Moreover, many states in this energy region share the same symmetry and are therefore coupled by this particular mode. We conclude that the observed dynamics correspond to cationic excited molecules that coherently vibrate. We note that the effect is observed only at the lowest electron energy (0.4 eV), which corresponds to the ionization out of 6ag orbital. Neither the calculations, nor the experiment show oscillations in the dynamics triggered by the ionization of the other two orbitals 5b1u (mapped to 0.64 eV signal) and 4b2u (mapped to 0.88 eV).”

“Experimental signal integrated over the electron kinetic energy in the range of 0.4±0.75 eV. The inset shows the FFT of the experimental signal taken from 50 to 250 fs (black) compared to the theory taken from 20 to 200 fs (green). This leads to an oscillation period of 65 fs.»

Reviewer #2 (Remarks to the Author):

1. The paper main focus is on the relaxation dynamics of multi-electron states in naphthalene and Adamantane. The relaxation timescales are inferred from decaying second (IR) ionization signal, but authors do not discuss the actual relaxation pathways and products. Does the decay simply refer to the downward transitions in the Franck-Condon region to some final level? Do these excited cations fragment? If so, how does the fragmentation impact the photoelectron yield?

The referee is correct. The decay mechanism refers to the transfer of population of excited cationic states towards lower in energy cationic states. We consider that the process is driven only by stable cationic molecules. This is supported by the fact that in the experiment the same time scale is obtained when measuring the time-dependent total electron signal and the time dependent dication signal (see SI Fig 4-e). Moreover, the time-dependent increase of the dicationic yield corresponds to a depletion of the cationic yield with the same time dependency, indicating that the excited cations eventually relax towards stable cations (we have included now these measurements in the SI Fig 4-d,e). The fact that the dynamics is observed in the electron, dication, and cation stable molecules seems to indicate that the dissociation plays a minor role in the effect measured. Moreover, we have observed the same dynamics in larger PAH where no fragmentation occurs (unpublished data).

In addition, we have also measured dynamics in the two-color fragment yield. The same short timescale dynamics is observed in the fragments, which can also be understood as the decay of excited cationic states. In that case the fragments correspond to cationic states excited to dissociative states by the IR probe pulse (shown in SI Fig 4-f,g).

2. Authors report the delay dependence of photoelectron yields in figure 2 as their main set of results, but do not provide the XUV harmonic spectrum used to obtain that data, except the fact that it extends from 17-35 eV. As harmonic 11 and 13 are below IP2 by 4.5 eV and 1.35 eV respectively, they cannot contribute to the 2 color electron yields reported in figure 2 at electron energies 0.35, 0.65, and 0.9 eV. The next harmonic (15th) is above the IP2 by 1.75 eV, and from figure S5 it seems to have a bandwidth of 0.3 eV. The IR dressing of direct photoelectrons can thus result in signals at $(1.75-1.55) \pm 0.15$ eV, i.e. between 0.05-0.35 eV. This contribution can enhance the 0.35 eV electron signals levels around time zero, which may manifest as a faster apparent decay. This process should not affect the analysis of photoelectrons at 0.65 and 0.9 eV. How do authors take into account of such IR dressing contributions?

We have added the spectra used for the experimental results in figure 2. We have also transferred from the SI to the main text the set of data showing that the time scale extracted from the 3 contributions remain unchanged (within the error bar) when the IR intensity and the XUV spectrum is varied (see Figure 2). We have added the text below.

“We have also changed the XUV spectrum and IR intensity and no significant variation of the timescale was observed (Fig 2.d,f)”

The referee is pointing out a very interesting effect considering that, at the XUV-IR overlap, the absorption accompanied by the emission of one IR photon will lead to a contribution at threshold photoelectron kinetic energy that might contribute to the short timescale measured around 0.4 eV.

We would expect that this effect will also vary when the XUV spectrum is changed (for instance when the 15th harmonic is small compared to higher ones) and when the IR intensity is altered. However, we have not observed any noticeable change of the timescale at threshold (see above). For every HH spectrum and IR intensity, we observed a smooth increase of the lifetime. Therefore, we conclude that the timescales reveal the intrinsic dynamics of the excited cations and even though such an effect might be present, our experiment is not sufficiently sensitive to observe it.

3. Other related experimental questions are: (a) Was the IR polarization parallel to the XUV? Did authors study the dependence of two-color electron yield on IR polarization, or

the angular distribution photoelectrons? (b) Authors mentioned they varied the intensity of IR probe. How does that effect the 2 color photoelectron signals? Did they discount the possibility of 2 IR photon transitions? (c) Are there any neutral Rydberg states below IP2 that could be excited by the XUV?

- (a) The IR polarization was indeed set parallel to the XUV polarization. This is now mentioned in the text and in Figure 1. We have studied the photoelectron angular distribution but have not found any noticeable effect within the measurement uncertainty. Therefore, no further information could be drawn from the angular distribution.
- (b) As shown in figure 2-f the IR intensity has been varied but no effect, apart from the increase of the total signal, has been observed. We, therefore, conclude that the IR serves only as a probe and does not contribute to other phenomena.
- (c) In principle, neutral Rydberg states below the double IP threshold might exist and, therefore, might get populated upon resonant XUV photoexcitation. Being already in the electronic continuum of the molecule these states will autoionize to low-lying cationic states. The electrons produced in this way, however, will have much higher energies than those of interest to us, as the molecule will mostly decay to the ground or the first few excited cationic states. In any case the contribution of these electrons is removed in our analysis by subtracting the “XUV only” signal. We also expect that the XUV excitation of Rydberg states with the many harmonics is less probable than the direct ionization.

4. In general, the observation of oscillations corresponding to vibrational coherences in figure 4 requires that the wave packet be accessible at specific coordinates or times, but not at others. What provides this selectivity in the experiment? Also from the inset, it seems that 1ag and 2ag are out-of-phase. In such a case, what determines the observed oscillation contrast?

Our calculations show that an oscillatory population transfer observed population oscillations appears along a specific vibrational mode (ag mode at 514.30 cm⁻¹) and two electronic states labeled 1ag and 2ag mostly take part in this dynamics. The referee correctly points out that in order to observe wavepacket oscillations the experiment should be probing the wave packet only at specific coordinates or times. In our experiment the selectivity came from the fact that the 2ag state belongs to the experimentally accessible energy region, whereas the 1ag state lies below it and thus is not probed. This means that the wave packet is only accessible at the time when it evolves on the 2ag state

A note clarifying this issue has been included in the main text of the article and in the caption of figure 4.

“Comparing to the experimental data, and taking into account the cross-correlation between the XUV and IR pulses, we see a similar recurrent signal (see Fig 4.b-c). The analysis shows that these recurrences correspond to coherent vibrational dynamics of the low-frequency ag mode at 514.3 cm⁻¹ (see inset of Fig 4.a) that couples two electronic states, one lying within (2ag) and one lying below (1ag) the energy region probed experimentally, making the wave packet experimentally accessible only during the time it evolves on 2ag surface. This mode is of particular importance, as it is the slowest totally symmetric mode of the molecule and thus easy to excite. Moreover, many states in this energy region share the same symmetry and are therefore coupled by this particular mode. We conclude that the observed dynamics correspond to cationic excited molecules that coherently vibrate. We note that the effect is observed only at the lowest electron energy (0.4 eV), which corresponds to the ionization out of 6ag orbital. Neither the calculations, nor the experiment show oscillations in the dynamics triggered by the

ionization of the other two orbitals 5b1u (mapped to 0.64 eV signal) and 4b2u (mapped to 0.88 eV).”

5. The IR pulse duration is listed as 25 fs, however, the XUV pulse duration is not explicitly mentioned. In reference to equation 1 of the supplement, I am also curious as to what is the value of τ_{crossco} extracted from higher energy features ?

The XUV pulse duration was not explicitly measured. We have extracted a cross-correlation of 40 fs from the high energy electrons, in very good agreement with the cross-correlation fitted as a free parameter at low electron energy. This is now mentioned in the text.

Overall, taken together with extensive supplementary information, the paper represents detailed effort on gaining new understanding of multi-electronic and non-BO effects in molecules. The experimental and theoretical results compliment each other. I would recommend publication the manuscript provided the above questions can be addressed and manuscript can be improved by transferring few pieces of vital information from the supplement to the manuscript.

We have transferred part of the SI to the main text as discussed in response to referee 1&2 as well as modified the figures.

Reviewers' comments:

Reviewer #1 (Remarks to the Author):

Marciniak et al revised their manuscript following the comments from reviewers.

The readability of their manuscript is significantly improved and the interpretation on the observed multielectronic and non-Born Oppenheimer ultrafast dynamics in naphthalene and adamantane is well described.

I can recommend that the content is worthy to be published as an article in the Nature Communications in the present form.

Reviewer #2 (Remarks to the Author):

The response of authors to referee 1, question 3 is confusing. Energy interval of +/- 0.75eV for integration is mentioned, and the revised manuscript and figure 3 caption also includes this number. This is a large area interval which will span various energy features. I thought the energy interval for integration was only 0.15 eV as per the original manuscript. Can the authors clarify what is the correct energy interval?

The work represents an interesting experimental and theoretical approach to extract multi-electronic and non-BO effects from an experimental data that likely contains many convoluted effects. In response to my question 2, authors invoke the lack of IR intensity dependence of timescales in figure 2(f) data. I am curious if the electron yields actually increase with IR intensity, or if the IR transitions are already saturated at 1 TW/cm². In the latter case, the fact that deduced timescales do not depend on IR intensity would not be a surprise. The lack of time-scale dependence on the strength of 15th harmonic is a good argument. To completely rule out the contribution from IR dressing it would be advisable to look at the presence of electron signals at $1.75+1.55 = 3.3$ eV, and this would help to set the upper limit for potential systematic error, if any.

In response to my question 3, authors agree that Rydberg states are a possibility. If so, then IR ionization from those states would contribute to the observed low energy electrons. For example, the harmonic below the IP2 could populate neutral states that will be ionized by IR to produce ~0.2-0.3 eV electrons. I wonder if this makes a significant difference to the theoretical analysis. I doubt

that it will, but an estimate of the excitation cross-section of the neutral states vs. the direct ionization cross-section would help to solidify the results.

By the way, there is no mention of figure 2(f) in the caption of figure 2.

While there are some limitations on how much information can be extracted from the data and modeling in such a complex system, the authors have made a good case and reported new insights. Overall the responses are satisfactory, and the paper could be published after the above mentioned points are clarified.

Reviewer #2 (Remarks to the Author):

The response of authors to referee 1, question 3 is confusing. Energy interval of +/- 0.75eV for integration is mentioned, and the revised manuscript and figure 3 caption also includes this number. This is a large area interval which will span various energy features. I thought the energy interval for integration was only 0.15 eV as per the original manuscript. Can the authors clarify what is the correct energy interval?

We apologize for this typo. The interval is +/- 0.075eV (not 0.75eV), which corresponds to the 0.15eV width indicated in our previous manuscript. We thank the referee for noticing this mistake, which is now corrected.

The work represents an interesting experimental and theoretical approach to extract multi-electronic and non-BO effects from an experimental data that likely contains many convoluted effects. In response to my question 2, authors invoke the lack of IR intensity dependence of timescales in figure 2(f) data. I am curious if the electron yields actually increase with IR intensity, or if the IR transitions are already saturated at 1 TW/cm². In the latter case, the fact that deduced timescales do not depend on IR intensity would not be a surprise. The lack of time-scale dependence on the strength of 15th harmonic is a good argument. To completely rule out the contribution from IR dressing it would be advisable to look at the presence of electron signals at $1.75+1.55 = 3.3$ eV, and the would help to set the upper limit for potential systematic error, if any.

In our experiment, the IR intensity could only be increased up to the point where above threshold multiphoton ionization (ATI) processes from the neutral molecules occur, since the ATI would produce low-energy photoelectrons that would affect the two-color photoelectron signal. Moreover, using too low IR intensity was also not possible due to the lack of signal. Therefore, the measurements have been performed in the range of IR intensity presented in figure 2.f. Within this intensity range, the two-color signal increases, but no detectable variation of the timescale was observed.

We think that IR dressing (XUV +/- IR photon) effects are negligible for the following reason: First, because doubling the IR intensity does not significantly change the observed timescale. Second, the dressing effects in the case of a complex system (with many close lying electronic states) will not lead to well-defined electron bands, like in atoms, but rather to a broad homogenous structure, especially at low electron energy, due to the contribution of the large number of states with different binding energies. This contribution is expected to be constant on the range of the studied photoelectrons and would lead to a global shift of the lifetime, not to the progressive increase observed.

The weak contribution of the IR dressing is also supported by Fig 2.d which shows that by changing the HHs spectra, no clear difference can be observed in the decay timescales.

In response to my question 3, authors agree that Rydberg states are a possibility. If so, then IR ionization from those states would contribute to the observed low energy electrons. For example, the harmonic below the IP2 could populate neutral states that will be ionized by IR to produce ~0.2-0.3 eV electrons. I wonder if this make a significant difference to the theoretical analysis. I doubt that it will, but an estimate of

the excitation cross-section of the neutral states vs. the direct ionization cross-section would help to solidify the results.

As we wrote in our previous reply to this question, neutral Rydberg states most likely exist in the energy range studied. These should be states corresponding to the excitation of an inner-valence electron to high-lying unoccupied molecular orbitals, and should autoionize very fast to low-lying cationic states. Indeed, before the autoionization takes place, the excited electron can be further ionized by the IR probe and thus in principle contribute to the signal. This contribution, however, should be negligible which is confirmed by the fact that the observed dynamics do not change with the change of the pump XUV spectrum (see Fig. 2d). Being a resonant process, the population of the Rydberg states in question will strongly depend on the XUV spectrum. Moreover, the overall dynamics and timescale observed in the electron signal also appears in the dicationic signal, meaning that it is correlated to the loss of 2 electrons by the neutral molecule. We conclude, therefore, that the pump pulse populates mainly cationic states in the energy region of interest like described in our model.

By the way, there is no mention of figure 2(f) in the caption of of figure 2.

We apologize for this oversight. This has been corrected in the new version.

While there are some limitations on how much information can be extracted from the data and modeling in such a complex system, the authors have made a good case and reported new insights. Overall the response are satisfactory, and the paper could be published after the above mentioned points are clarified.

We thank the referee for his/her valuable input on our manuscript, which has helped us to improve the discussion of the results.

REVIEWERS' COMMENTS:

Reviewer #2 (Remarks to the Author):

After the second round of revisions and clarifications, the paper is now acceptable for publication.